# Extracellular Vesicles from *Leishmania (Leishmania) infantum* Contribute in Stimulating Immune Response and Immunosuppression in Hosts with Visceral Leishmaniasis

**DOI:** 10.3390/microorganisms12020270

**Published:** 2024-01-27

**Authors:** Francieli Marinho Carneiro, Allecineia Bispo da Cruz, Marta Marques Maia, Noemi Nosomi Taniwaki, Ingrid de Siqueira Pereira, Gislene Mitsue Namiyama, Ricardo Gava, Roberto Mitsuyoshi Hiramoto, Bruno Vicente, Victor Midlej, Rafael Meyer Mariante, Vera Lucia Pereira-Chioccola

**Affiliations:** 1Centro de Parasitologia e Micologia, Instituto Adolfo Lutz, Sao Paulo 01246-000, Brazil; francielimarinho@live.com (F.M.C.); abc_alle@hotmail.com (A.B.d.C.); m.marquesmaia.3@gmail.com (M.M.M.); siqueira_ingrid@hotmail.com (I.d.S.P.); gava44@gmail.com (R.G.); roberto.hiramoto@ial.sp.gov.br (R.M.H.); 2Núcleo de Microscopia Eletrônica, Instituto Adolfo Lutz, Sao Paulo 01246-000, Brazil; ntaniwak@hotmail.com (N.N.T.); ginamiyama@yahoo.com.br (G.M.N.); 3Laboratório de Biologia Estrutural, Fundação Oswaldo Cruz, Instituto Oswaldo Cruz, Rio de Janeiro 21040-900, Brazil; bruno-jpa1@hotmail.com (B.V.); victor.midlej@ioc.fiocruz.br (V.M.); rafaelmariante@gmail.com (R.M.M.)

**Keywords:** extracellular vesicles, *Leishmania (Leishmania) infantum*, THP-1 cells, cytokines, miRNAs

## Abstract

Visceral leishmaniasis (VL) is a chronic systemic disease. In Brazil this infection is caused by *Leishmania (Leishmania) infantum*. Extracellular vesicles (EVs) released by *Leishmania* species have different functions like the modulation of host immune systems and inflammatory responses, among others. This study evaluated the participation of EVs from *L. (L.) infantum* (Leish-EVs) in recognition of the humoral and cellular immune response of hosts with VL. Promastigotes were cultivated in 199 medium and, in the log phase of growth, they were centrifuged, washed, resus-pended in RPMI medium, and incubated for 2 to 24 h, at 25 °C or 37 °C to release Leish-EVs. This dynamic was evaluated using transmission (TEM) and scanning (SEM) electron microscopies, as well as nanoparticle tracking analysis (NTA). The results suggested that parasite penetration in mammal macrophages requires more Leish-EVs than those living in insect vectors, since promastigotes incubated at 37 °C released more Leish-EVs than those incubated at 25 °C. Infected THP-1 cells produced high EV concentration (THP-1 cells-EVs) when compared with those from the control group. The same results were obtained when THP-1 cells were treated with Leish-EVs or a crude *Leishmania* antigen. These data indicated that host–EV concentrations could be used to distinguish infected from uninfected hosts. THP-1 cells treated with Leish-EVs expressed more IL-12 than control THP-1 cells, but were unable to express IFN-γ. These same cells highly expressed IL-10, which inhibited TNF-α and IL-6. Equally, THP-1 cells treated with Leish-EVs up-expressed miR-21-5p and miR-146a-5p. In conclusion, THP-1 cells treated with Leish-EVs highly expressed miR-21-5p and miR-146a-5p and caused the dysregulation of IL-10. Indirectly, these results suggest that high expression of these miRNAs species is caused by Leish-EVs. Consequently, this molecular via can contribute to immunosuppression causing enhanced immunopathology in infected hosts.

## 1. Introduction

Visceral leishmaniasis (VL) is caused by protozoans classified in *Leishmania (Leishmania) donovani* complex. They are transmitted by infected phlebotomine sandflies (Diptera: Psychodidae). VL is by far the most severe form of the leishmaniases, since it compromises internal organs and is highly lethal for untreated hosts. Severe infection affects humans and domestic dogs in peridomestic and domestic transmission foci, and is included in the neglected diseases. If adequate treatment is not initiated in a timely manner, infected hosts progress to death in 90% of cases [1,2,3,4,5,6,7].

Globally, VL is among the top-ten neglected tropical diseases, with more than 12 million people and around 1 billion people at risk of infection [7]. This disease has spread worldwide and, at the present time, is found in the Americas, Africa, Southern Europe, and Asia. The majority of cases occur in India, Bangladesh, Sudan, Nepal, and Brazil [1,3,7,8]. Around 90% of the cases in Latin America occur in Brazil. VL in Brazil is caused by *Leishmania (Leishmania) infantum*, transmitted by the main vector, *Lutzomyia longipalpis* [8,9,10,11]. Currently, human and canine infections are present in wild and rural environments as well as in urban centers. Unfortunately, the control strategies against the disease, vectors, and reservoirs of VL have been ineffective [1,2,3,4]

The life cycle of *L. (L.) infantum* is heteroxenic with two main life forms. Promastigotes, which live in infected female sandflies and are regurgitated during the blood meal, and amastigotes, which live inside the phagocytic cells of the vertebrate host [2,8]. In the beginning of infection, promastigotes interact with a complex extracellular matrix, immune system molecules, and phagocytic tissue cells. Next, macrophages eliminate the phagocytosed parasites by production of microbicidal molecules, and they stimulate the effector immune response to release immunoregulatory and proinflammatory cytokines, such as TNF-α, IL-10, and IL-6 [12,13].

Despite the complex host system, all *Leishmania* species have several virulence factors that are capable of surviving the macrophage effector response, causing Leishmaniasis pathogenesis [14,15]. In order to guarantee the invasion in the host cells, pathogens release effector molecules through the cellular membrane, and the via extracellular vesicles (EVs) are transported to host cells. These compounds cause the modulation of cytokines and microRNAs (miRNAs) for preparing the cells for parasite infection [16,17,18,19]. In vivo studies, analyzing mice and macrophages, complement these findings, reporting that EVs released by *Leishmania* (Leish-EVs) can modulate the immune-regulating and signaling pathways [19,20,21,22,23,24].

EVs represent all particle populations. The largest are the apoptotic bodies that arise as blogging from the apoptotic cell membranes. They are phosphatidylserine-rich vesicles with 500–5000 nm in diameter. The microvesicles are particles of 100–1000 nm in diameter that shed from the plasma membrane. They are enriched with phosphatidylserine and cholesterol. The exosomes are the smallest EVs (30–150 nm) and are formed by the exocytosis of multivesicular bodies (MVBs). They are liberated intraluminal vesicles upon fusion with the plasma membrane [25,26,27,28]. Studies have shown that the different cell types use EVs as vehicles for delivery of modulatory proteins, lipids, and nucleic acids to other short- and long-distance cells. In addition, EVs are critical mediators of intercellular communication potentiated organ cross-talk [27,28,29,30,31,32].

miRNAs are small endogenous non-coding RNAs. They are produced by all cells and a part of them is transported by EVs to biological fluids to promote different biological functions. Each miRNA can target hundreds of mRNAs within a given cell type, and a single mRNA is often the target of multiple miRNAs [33,34]. Each molecule contains around 22 nucleotides and post-transcriptionally regulates the expression of several tar-gets [35]. Studies have been shown that miRNAs are promising biomarkers for many hu-man diseases, and they can be investigated through non-invasive methods. Furthermore, miRNAs are able to assess the prognosis of a disease and the response to treatment [36].

Based on these findings, the goal of this study was to evaluate the participation of Leish-EVs, released by *L. (L.) infantum*, in the recognition of the humoral and cellular immune response of hosts with VL. The investigations described throughout this article in-directly suggest that Leish-EVs trigger in infected hosts a molecular via that can contribute to the immunosuppression and production of cytokines.

## 2. Methods

### 2.1. Ethical Statements

This study was performed in accordance with the recommendations of the Ethics Committee for Animal Use of the Adolfo Lutz Institute (Comissão de Ética no Uso de Animais do Instituto Adolfo Lutz, CEUA-IAL) and the National Experimentation Control Council (Conselho Nacional de Controle de Experimentação, CONCEA). Both committees approved this study (Protocol: A-CEUA-003).

### 2.2. L. (L.) infantum Cultures and Production of Crude Leishmania Antigen (CLA) and Leish-EVs

The protocols were prepared as described before [19]. Promastigotes of *L. (L.) infantum* (MHOM/BR/1972/LD) were maintained at 25 °C, in 199 medium, pH 7.2 containing fetal bovine serum (FBS) 10%, gentamicin 30 μg/mL, hemin 50 μL/mL, and human urine 50 μL/mL.

For use in experiments, promastigotes were collected from culture medium in the log phase of growth, counted, washed 3 times, and concentrated through centrifugation (3000× *g* for 10 min) with sterile phosphate-buffered saline (PBS).

For CLA preparation, promastigotes dissolved in sterile in PBS (2 mL) were ruptured in Disruptor (Tissue Lyser, Qiagen, Hilden, Germany) through agitation by 8 cycles of 4 min (50 oscillations/second) with 2 min intervals between cycles and sonicated for 10 min. After certifying the lysis of the parasites through optical microscopy, the CLA was dissolved in 0.3 M NaCl.

For Leish-EVs, promastigotes were concentrated through centrifugation (2000× *g* for 15 min) and washed (5 times) with sterile PBS. Next, parasites were resuspended in RPMI medium (1 mL) and incubated for 2 h at 25 °C to release vesicles in the culture medium. The mixture, containing promastigotes and Leish-EVs, was filtered (0.22 μm filter) to remove the parasites.

The protein concentrations of Leish-EVs and CLA were estimated using BCA protein kit (Pierce) according to the manufacturer’s instructions and by a NanoDrop spectrophotometer (Thermo Fisher Scientific, Waltham, MA, USA) at 280 nm. Next, aliquots were treated with 10 μg/mL of a cocktail of protease inhibitors containing, per mL, 20 µM AEBSF (4-(2-Aminoethyl) benzenesulfonyl fluoride hydrochloride), 10 µM EDTA, 1.3 µM Bestatin, 0.14 µM E-64, 10 nM Leupeptin, and 3 nM Aprotinin (Sigma-Aldrich, St. Louis, MO, USA). Thereafter, each aliquot was stored at –20 °C until analysis.

For immunological reactions, the supernatants containing Leish-EVs were transferred to Ultra-Clear centrifuge tubes (6 mL tube for SW-55 rotor), (Beckman Coulter, Brea, CA, USA) and the volume (6 mL) was completed with filtered PBS. Samples were ultracentrifuged at 100,000× *g* for 60 min at 25 °C in a Beckman^®^ Coulter L8-80M centrifuge. The pellet containing Leish-EVs was suspended in 100 µL of filtered PBS. For ultrastructural analyses, promastigotes were analyzed in 5 moments with incubation times of T0, T2, T4, T6, and T24 h.

### 2.3. Ultrastructural Analyses of Promastigotes Releasing Leish-EVs through Transmission Electron Microscopy (TEM) and Scanning Electron Microscopy (SEM)

For ultrastructural analyses in TEM and SEM of promastigotes releasing Leish-EVs, 5 sets of parasites were analyzed based on the incubation time for EV release at temperatures of 25 °C or 37 °C. They were: 0 (in the beginning), 2, 4, 6, and 24 h.

For analysis of Leish-EVs through TEM, supernatants containing Leish-EVs were transferred to Ultra-Clear centrifuge tubes (6 mL tube for SW-55 rotor, Beckman Coulter, Brea, CA, USA), and the volume was completed to 6 mL with filtered PBS. Tubes were ultracentrifuged at 100,000× *g* for 60 min at 25 °C in a Beckman Coulter L8-80M centrifuge. Next, pellets containing Leish-EVs were suspended in filtered PBS (100 µL).

Promastigotes releasing Leish-EVs or only Leish-EVs were fixed with 2.5% glutaraldehyde, 4% paraformaldehyde in 0.1 M sodium cacodylate buffer, pH 7.4, for at least 2 h. Then, pellets were rinsed in the same buffer and post-fixed in a solution containing 1% osmium tetroxide, 0.8% ferrocyanide, and 5 mM calcium chloride; washed in 0.1 M sodium cacodylate buffer; dehydrated in graded acetone; and embedded in Epon resin. Ultrathin sections (100 nm) were obtained in a Sorvall ultramicrotome; stained with uranyl acetate and lead citrate; and observed under a Transmission Electron Microscope JEOL (model JEM 1011, Peabody, MA, USA) operating at 80 kV. Images were captured on a charge-coupled device camera (model 785 ES1000W, Gatan, Pleasanton, CA, USA) and the Gatan version 1.6 programs.

For high-resolution SEM analysis, after incubations, 1 × 10^6^/mL promastigotes were washed twice in filtered PBS and adhered on coverslips previously coated with 0.1% aqueous poly-L-lysine (Sigma) for at least 2 h at room temperature. Next, coverslips were fixed in 2.5% glutaraldehyde in 0.1 mol/L sodium cacodylate buffer, pH 7.2, containing 0.146 mol/L sucrose and 5 mmol/L CaCl_2_ for 1 h at room temperature. The parasites were post-fixed in 1% osmium tetroxide (OsO_4_) for 15 min and then dehydrated in increasing concentration series of ethanol, 7.5%, 15%, 30%, 50%, 70%, 90%, and 100% (*v*/*v*), for 15 min in each step. Next, the samples were critical point-dried with CO_2_ and sputter-coated with a thin layer (2 nm) of platinum. The images were recorded at accelerating voltages of 1 Kv on microscopes equipped with the Field Emission Gun (FEG): a Quattro ESEM (Thermo Fisher, Waltham, MA, USA) or an Auriga High-Resolution SEM (Zeiss, Jena, Germany).

### 2.4. THP-1 Cell Line and THP-1 Released EVs (THP-1-EVs) Production

The cell line used in experiments was the THP-1 cell line (ATCC nº TIB 202–The Global Bioresource Center, Manassas, VA, USA), a human pro-monocyte cell line derived from an acute monocytic leukemia patient. THP-1 cell line was maintained in culture bottles containing RPMI medium, 20% FBS, 2 mM glutamine, 10% gentamicin, at 37 °C in 5% CO_2_. For experiments, THP-1 cells (5 × 10^5^ cells/well) were plated, in duplicate, in 24-well plates containing the same medium, stimulated and differentiated by Phorbol 12-Myristate 13-Acetate (PMA) 160 ng/mL, and non-adherent cells were removed after macrophage adhesion.

After the multiplicity of infection (MOI) calculation, macrophages were infected with promastigotes at proportion 5:1. In parallel, THP-1-cells were treated with CLA or Leish-EVs at concentration 20 µg/well.

For each experiment, a normal control was included, which constituted the same macrophage concentration but without infection or treatment. THP-1 cells were incubated at 37 °C for 24 h, in 5% CO_2_ to release THP-1-EVs. All experiments were performed in duplicate. The supernatants were collected (500 μL/well) without any treatment. Next, macrophages and supernatants were kept at −70 °C until experiments. Subsequently, supernatants were used for determination of THP-1-EV concentrations by NTA, and miRNAs expression. The macrophages were used for cytokine determinations.

### 2.5. Concentration Determination of Leish-EVs and THP-1-EVs through Nanoparticle Tracking Analysis (NTA)

Each experiment was performed twice to confirm the results. Leish-EVs and THP-1-EVs were evaluated for concentration/mL through nanoparticle tracking analysis (NTA) on the NanoSight NS300 equipment (Malvern- NanoSight™, NTA 3.0), coupled to an automatic sCMOS camera with a wavelength of 532 nm. NanoSight computes size and particle number based on the measured Brownian motion. The coefficient and hydrodynamic radius were determined using the Stokes–Einstein equation and the results were exhibited as particle size distribution. Captures and analyses were performed according to the equipment protocol. Data were shown as the average and standard deviation of three video recordings of 30 s per sample. Since NTA is accurate between particle concentrations in the range of 2 × 10^7^ to 2 × 10^9^/mL, the aliquots (EVs) were diluted before analysis in filtered PBS that was used as a control. Next the relative concentration was calculated according to dilution factor.

### 2.6. Immunological Investigations

The experiments were carried out to investigate whether Leish-EVs could recognize the humoral response in infected hosts. For this purpose, five serum samples from dogs with canine visceral leishmaniasis (Can-VL) were tested by Immunoblotting and Dot blot, using Leish-EVs as antigen and CLA, as a positive antigen control. Five healthy dogs were used for negative control. The canine sera have been previously tested for Can-VL through ELISA, Fast test, and real-time PCR [19,37].

For immunoblotting, CLA or Leish-EVs (20 µg/mL for both) were solubilized in lysis buffer (2% SDS, 10% glycerol, 5% 2-mercaptoethanol, 60 mM Tris-HCl, pH 6.8, and 0.002% bromophenol blue), boiled, and run in 12% polyacrylamide-SDS gels. After visualization through silver staining, Leish-EVs and CLA proteins were transferred to nitrocellulose membranes, blocked with 5% skim milk-PBS (at room temperature/60 min), and washed with PBS (5 times/5 min). Membranes containing proteins were incubated for 18 h (4 °C) with sera (diluted 1:50 in PBS-5% skimmed milk). After another five washes and drying phases (around 2 h), the membranes were incubated at room temperature, under agitation (2 h) with a peroxidase-conjugated rabbit anti-dog IgG (diluted 1:1500 in 5% skim milk-PBS) and washed again.

For the Dot blots, strips of nitrocellulose membranes (0.45 μm in diameter) were incubated with CLA or Leish-EVs (1.5 μg/mL). After five washes with PBS (5 min), the strips were incubated for 18 h (4 °C) with each serum (diluted 1:50 in 5% skim milk-PBS). After five more washes with PBS (5 min) and drying (at least 2 h), strips were incubated at room temperature, under agitation (2 h) with peroxidase-conjugated rabbit anti-dog IgG (diluted 1:4000 in 5% skim milk-PBS) and washed again.

Bound antibodies from Dot blots and immunoblotting were visualized after treatment with chemiluminescence Western blotting substrate (Pierce ECL Western Solution, Thermo Scientific, Waltham, MA, USA). The images were captured on documenting gel with a chemiluminescence filter (UVITEC, Cleaver Scientific, Rugby, UK) in a Blot Scanner (C-DiGit).

### 2.7. Cytokine and miRNA Analysis by Gene Expression in Quantitative Real-Time PCR (qPCR)

To evaluate whether Leish-EVs could stimulate the cellular immune response, THP-1 cells (5 × 10^5^ cells/well) were incubated for 24 h with Leish-EVs (20 µg/mL). For each experiment, a normal control was included, which constituted the same cell concentration but without treatment. Prior to the experimental process and handling samples, all materials and working surfaces were cleaned with RNase decontamination solution (RNaseZap—Thermo Fisher) in order to minimize the RNA degradation.

Reactions for cytokine and miRNA expression were performed as described before [38,39]. For cytokine investigation, RNA molecules were extracted from THP-1 cells using RNeasy Mini Kit (Qiagen) according to the manufacturer’s instructions. Isolated RNA molecules were treated with RNA later to remove residual genomic DNA. Next, 10 μL of each sample was reverse-transcribed (RT) using a high-capacity cDNA reverse transcription kit in a Veriti 96-Well Thermal Cycler, according to the manufacturer’s instructions under the following thermal conditions: 10 min at 25 °C, 120 min at 37 °C, followed by 5 min at 85 °C. cDNA samples were stored at −70 °C until use in qPCR. Each qPCR amplification mixture contained 5 μL of 2× TaqMan Universal PCR Master Mix and 0.5 μL TaqMan Gene Expression Assays were mixture with the following genes: Interferon gamma (IFN-γ), Interleukin 10 (IL-10), Interleukin 12 (IL-12), Interleukin 6 (IL-6), Transforming growth factor beta (TGF-β), Tumor necrosis factor-alpha (TNF-α), and glyceraldehyde-3-phosphate dehydrogenase (GAPDH). Assay IDs are shown in Table 1. Template cDNA (2 µL) and 2.5 µL of RNAse-free water were added to a total volume of 10 µL. Reactions were prepared in duplicate for each sample and all assays. Samples were amplified and detected using a StepOne Real-Time PCR System (Applied Biosystems, Waltham, MA, USA) using the following thermal profile: 2 min, 50 °C, and 95 °C for 10 min, followed by 40 cycles performed at 95 °C for 15 s and 60 °C for 1 min. The results with no amplification in qPCR were repeated at least two times.

In parallel, miRNA expression profiles were assessed in supernatants of the same samples using the miRNeasy Serum/Plasma Kit (Qiagen, Valencia, CA, USA) and miRNAs were eluted into 30 µL of nuclease-free water. The normalization of miRNA expression started in miRNA extractions, since each sample was spiked with 5 µL (25 fmol) of the spike-in control (*Caenorhabditis elegans* miRNA, the synthetic molecule cel-miR-39, Ambion). Therefore, cel-miR-39-3p was chosen as an exogenous gene. The success of the extractions was confirmed through qPCR amplification of the exogenous miRNA. For cDNA synthesis, 2 μL of total RNA was used in an RT-PCR using TaqMan Advanced miRNA cDNA synthesis kit (Applied Biosystems). RT was performed according to manufacturer’s instructions, in four steps, under the following thermal conditions: 45 min at 37 °C, 10 min at 65 °C for poly (A) tailing reaction; 60 min at 16 °C for ligation reaction; 15 min at 42 °C, 5 min at 85 °C for reverse transcription reaction; 5 min at 95 °C, followed by 14 cycles at 95 °C for 3 s and 60 °C for 30 s; stop reaction at 99 °C for 10 min for miR-Amp reaction. After cDNA synthesis, samples were diluted 1:10 according to the manufacturer’s instructions. qPCR was performed in a customized assay produced by Applied Biosystems. Each qPCR amplification mix contained TaqMan Fast Advanced Master Mix (5 µL), cDNA (2.5 µL), RNAse-free water (2 µL), and TaqMan Advanced miRNA Assays (0.5 µL) for each gene: miR-21-5p, miR-146a-5p, miR-125b-5p, and miR-144-3p. Table 2 describes, in detail, the characterization of each miRNA species. Reactions were performed in duplicate in a final volume of 10 µL and included a negative control (amplification mix only). The qPCRs were performed in the StepOne Real-Time PCR Systems equipment (Applied Biosystems) using the cycling program in Fast mode: one cycle of 95 °C at 20 s, 40 cycles of 95 C at 1 s, and 60 °C for 20 s.

### 2.8. Data Analysis

Statistical analyses and graphics were performed using Graph Pad Prism software version 8.4.3 (San Diego, CA, USA). Comparisons of concentration and size of Leish-EVs and THP-1-EVs were determined through an unpaired one-tailed test based on a critical value of *p* ≤ 0.05. Values are presented as mean ± SEM (standard error of the mean). The indicated data were representative of at least two individual experiments.

For cytokine and miRNA expressions, the data were based in amplification plot, which represented the cycle threshold (CT) values for each sample. The values of expression (cytokines and miRNAs) are showed as relative quantification (RQ) and were calculated through the comparative CT method (2^−ΔΔCT^) as described before [40]. THP-1 cells without infection or treatment (control) were considerate calibrators, which values were always 1.0 according to the comparative CT method. The use of a calibrator is recommended to describe how many times the studied gene is expressed (more or less than the calibrator).

In order to carry out this formula, the variation in cDNA concentration between samples (ΔCT value) was corrected by subtracting the CT value obtained from the exogenous gene (cel-miR-39-3p) or gene endogenous for cytokines (GAPDH). Next, ΔCT values from the tested samples were subtracted from the ΔCT value of the calibrator.

## 3. Results

The initial experiments were performed to define the ideal in vitro conditions for production of Leish-EVs and THP-1-EVs: promastigote number for each cell (MOI), ideal temperature, incubation time, and transformation of THP-1 monocytes in macrophages.

### 3.1. Biological Characteristics of Leish-EVs Released by L. (L.) infantum

The aim of these experiments was to investigate how *L. (L.) infantum* promastigotes release EVs over time in different environments. Figure 1 shows images captured by TEM and SEM of promastigotes releasing Leish-EVs, at 25 °C (vector temperature), in five periods. Figure 1A–F show promastigotes at the onset of Leish-EV releasing (T = 0). The images show the Leish-EVs inside the promastigotes and preparing to be released by the plasma membrane. After 2 h (Figure 1B,G), 4 h (Figure 1C,H), 6 h (Figure 1D,I), and 24 h (Figure 1E,J) incubation, it was possible to note different moments of Leish-EV release through the plasma membrane and those already released to the external environment. Figure 2 also shows images captured through TEM and SEM in the same five periods of promastigotes releasing Leish-EVs, but at 37 °C (mammal body temperature). Despite the fact that these images do not clearly show that promastigotes released more Leish-EVs than those promastigotes from Figure 1, in the NTA results (concentration of Leish-EVs/mL) allow us to note that promastigotes incubated at 37 °C released more Leish-EVs than promastigotes incubated at 25 °C (Figure 3A). However, the mean sizes of Leish-EVs were similar in both incubation periods and the majority of them had a size which was compatible with that of microvesicles (Figure 3B).

Promastigotes incubated in RPMI medium at 25 °C for 2, 4, 6, and 24 h released the mean concentrations (and sizes) of 2.54 × 10^8^ (150.8 nm), 7.15 × 10^8^ (195.9 nm), 9.45 × 10^8^ (205.5 nm), and 2.45 × 10^9^ (142.3 nm) Leish-EVs/mL, respectively. For those incubated at 37 °C, the mean concentrations (and sizes) were 1.75 × 10^9^ (142.3), 6.71 × 10^9^ (170.5), 1.71 × 10^10^ (198.4 nm), and 1.71 × 10^10^ (150.8 nm) Leish-EVs/mL, respectively.

### 3.2. THP-1-EVs Releasing Was Stimulated by L. (L.) infantum Antigens and Leish-EVs

The intention of these experiments was to define whether non-infected and infected macrophages with promastigotes can release the same amount of THP-1-EVs. For this purpose, THP-1 cells were infected with promastigotes at proportion 5:1 and incubated for 6 and 24 h. Next, the concentrations of THP-1-EVs in the supernatants were analyzed using NTA. The values represent the means of three reads in NanoSight equipment and the mean concentration ± SEM (standard error of the mean) of each aliquot, prepared in duplicate. The comparison between aliquots was statistically determined through unpaired, one-tailed t-tests. As shown in Figure 4, the mean concentration released by control aliquot (THP-1) was 1.6 × 10^9^ ± 1.9 × 10^8^ THP-1-EVs/mL. The mean concentration released by THP-1 aliquot, incubated with promastigotes for 6 h, increased to 2.7 × 10^9^ ± 9.5 × 10^7^ THP-1-EVs/mL. THP-1 cells incubated with promastigotes for 24 h released 3.8 × 10^9^ ± 1.5 × 10^8^ THP-1-EVs/mL. The THP-1 cells without infection (control) released, during 24 h, 1.8 × 10^9^ ± 3.8 × 10^8^ THP-1-EVs/mL. The increase in THP-1-EV production in aliquots from infected THP-1 cells (incubations with 6 and 24 h) compared with control aliquots was statistically different at * *p* < 0.018 and * *p* < 0.042, respectively.

In parallel, THP-1 cells were treated with CLA (20 µg/well) and were incubated for 6 or 24 h. This experiment was performed to understand whether the excess of EVs was released by promastigotes or not. The mean concentration released by control aliquot and incubated for 6 h was 1.0 × 10^9^ ± 3.4 × 10^8^ THP-1-EVs/mL. The mean concentration released by THP-1 aliquots treated with CLA increased to 5.6 × 10^9^ ± 3.8 × 10^8^ THP-1-EVs/mL. THP-1 cells incubated at 24 h released 1.5 × 10^9^ ± 7.0 × 10^7^ THP-1-EVs/mL (control). Those treated with CLA and incubated for 24 h released 8.2 × 10^9^ ± 1.0 × 10^9^ THP-1-EVs/mL. The increase in THP-1-EV production compared with both control aliquots was statistically different at * *p* < 0.022 and ** *p* < 0.006, respectively.

Next, THP-1 cells were treated with Leish-EVs at a concentration of 20 µg/well and incubated for 6 h or 24 h. The mean concentration released by the control aliquot incubated for 6 h was 1.2 × 10^9^ ± 9.0 × 10^7^ THP-1-EVs/mL. The mean concentration of THP-1 aliquots treated with Leish-EVs was 3.1 × 10^9^ ± 2.9 × 10^7^ THP-1-EVs/mL. For those without treatment (controls) incubated for 24 h, the mean concentration was 1.5 × 10^9^ ± 4.0 × 10^8^ THP-1-EVs/mL. In THP-1 aliquots treated with Leish-EVs, the mean concentration was 3.4 × 10^9^ ± 6.2 × 10^8^ THP-1-EVs/mL. The increase in THP-1-EV production compared with both control aliquots was statistically different at ** *p* < 0.018 and ** *p* < 0.051, respectively.

### 3.3. Immunological Experiments

The aim of the immunological assays was to evaluate whether Leish-EVs were recognized by sera from infected hosts. The assays were performed using Leish-EVs as antigen. CLA was used as a positive antigen control. Figure 5A shows the electrophoretic profile Leish-EVs on 12% SDS-PAGE after silver nitrate staining (strip 1). Next, Leish-EVs were analyzed as antigen in immunoblot using 2 canine sera. One was positive (strip 2) and the other was negative (strip 3) for Can-VL. As positive antigen control, CLA was used for the same sera for Leish-EVs, strips 4 and 5, respectively. Leish-EVs and CLA were recognized by the positive sera (strips 2 and 4) and were not recognized by negative sera (strips 3 and 5).

Equally, Figure 5B shows the same results of Dot blot using Leish-EVs as antigen and five positive and five negative canine sera for Can-VL. CLA was used as the antigen control.

### 3.4. Leish-EVs Stimulated the THP-1 Cells to Produce Cytokines and miRNAs

The results of the gene expression of cytokines produced by THP-1 cells after treatment with Leish-EVs are shown in Figure 6A.

IL-10 and IL-12 were expressed in the THP-1 cells treated with Leish-EVs by around 7 and 4 times more than the THP-1 cells without treatment (control), respectively. TGF-β, TNF-α and IL-6 also were up-expressed, although with low intensity. The values were 1.51, 1.20, and 1.11 times more expressed than controls (THP-1 cells without treatment), respectively. IFN-γ was not expressed in the tested samples.

The relative quantifications of the four miRNA species (miR-21-5p, miR-146a-5p, miR-125b-5p, miR-144-3p) were estimated from miRNAs purified from THP-1 cells (5 × 10^5^ cells/well) treated with Leish-EVs (20 µg) for 24 h. THP-1 cells without Leish-EV treatment were considered calibrators with a fixed value of 1.0. Thus, the values of treated cells inform us of how many times they were more expressed. miR-21-5p and miR-146a-5p were up-expressed in THP-1 cells treated with Leish-EVs (RQ means of 6.97 and 2.73, respectively). These values represent around 7 and 3 times more expression than the THP-1 cells without treatment (control), respectively. miR-125b-5p and miR-144-3p were slightly expressed, with RQ mean of 1.40 and 0.42, respectively. Figure 6B shows the RQ mean of each miRNA species.

## 4. Discussion

Diverse studies related with EVs and pathogenic protozoa interactions have shown that these nanoparticles play an important role in parasite survival, facilitating their infection in experimental models [41] and adapting the parasite to the host environment [42]. EVs carry virulence factors that impact the modulation of the cellular immune response, as has been shown in *Leishmania* species, such as *L. (L.) mexicana* [24], *L. (L.) major* [43], *L. (L.) amazonensis* [33], and *L. (L.) donovani* [34].

Similar to other *Leishmania* species, the images captured through electronic microscopy and NTA analyses suggest that *L. (L.) infantum* promastigotes released more Leish-EVs at 37 °C than those incubated at 25 °C. These results suggest that parasite penetration in mammal macrophages require more Leish-EV than those living in insect vectors. Probably, the virulence factors carried by Leish-EVs contribute to parasite adaptation and maintenance in host cells. These data are in concordance with previous studies [20,35]. Regarding Leish-EVs, the structures shown in NTA and TEM had the size and shape characteristics of microvesicles.

The macrophages play a dual role during infections caused by the different *Leishmania* species. They are responsible for internalizing and destroying parasites, and, at the same time, macrophages are safe places for promastigote multiplication. Thus, these cells are the key to the failure or success of the infection [44].

The experiments using THP-1 cells were performed to understand the relationship between the release of macrophages *L. (L.) infantum* and THP-1-EV during infection. The first investigations revealed that THP-1 cells infected with promastigotes were capable of producing an increase in the release of THP-1-EVs. The results demonstrated an increase in the concentrations of THP-1-EVs released from 6 to 24 h. Interestingly, THP-1 cells treated with the CLA or Leish-EVs had the same results. Similar results were observed in EVs from bone marrow in *L. (L.) amazonensis* [45]. The comparison between groups (control versus infected) were statistically different. These data indicate that the concentration of EVs may be able to distinguish infected from uninfected hosts [19].

The immunological analyses were performed to investigate the ability of the immune system from infected hosts to recognize the Leish-EVs. The immunoblots and Dot blots used in this study indicated a high capacity of Leish-EVs to be used as biomarkers in serological diagnosis of the Can-VL. Similar results have been seen in patients with VL (manuscript in preparation). These data confirm the findings of our previous study [19], in which we showed that Leish-EVs were highly antigenic for infected hosts (dogs with Can-VL).

In natural Can-VL infection, the protective cellular immunity is related to increase in Th1 mediated by IFN-γ and TNF-α. Thus, the macrophage activation and cytotoxic T lymphocytes exhibit apparent resistance to infection. In vitro studies have demonstrated that IFN-γ plays an essential role in combating infectious diseases. Induction of IFN-γ secretion by T and natural killer cells through synergistic stimulation with IL-12 and IL-18 in the adaptive immune responses against pathogens is well established. In the same way, naturally activated macrophages with treatment with IL-12 and IL-18 immediately secreted IFN-γ [46].

In symptomatic infection, the roles of IL-4 and IL-10 advance, reducing the effects of Th1 cytokines. In this case, the production of nitric oxide in macrophages decreases and prevents the destruction of the parasite [44]. Here, THP-1 cells treated with Leish-EVs expressed four times more IL-12 than the THP-1 cells without treatment, but they were unable to produce IFN-γ.

Leish-EVs are released to extracellular cargo, targeting the host cells; consequently, they cause immunosuppression in these cells to facilitate the parasite invasion [47].

IL-10 is an anti-inflammatory cytokine that prevents damage to the host. However, the dysregulation of IL-10 is associated with enhanced immunopathology in response to infection as well as increased risk for development of autoimmune diseases [48]. A previous study demonstrated that patients with VL had high IL-10 production that blocked the production of IL-6 and TNF α [49]. THP-1 cells treated with Leish-EVs expressed around four times more IL-12 than normal THP-1 cells, but they were unable to produce IFN-γ. However, these same cells highly expressed IL-10; this expression was around seven times higher than that of normal macrophages. At the same time, they inhibited TNF-α and IL-6. Although TGF-β had low expression, it cooperated with IL-10 in causing immunosuppression [44,45,46,47].

One of the functions of miRNAs is to directly activate cytokine production, regulating functions correlated with cellular immune responses [50,51,52,53]. Thus, the expression of miRNAs was analyzed to determine whether Leish-EVs could stimulate the expression of some miRNAs in macrophages. miR-21-5p and miR-146a-5p were up-expressed in THP-1 cells treated with Leish-EVs. Previous studies demonstrated that up-expression of miR-21-5p in tissue samples from dogs with Can-VL promoted an anti-inflammatory reaction, reducing the Th1 immune response among T cells and phagocytes [54]. In humans, the up-expression of miR-21-5p caused an imbalance between Th1 and Th2 responses [55,56,57,58]. Equally, miR-146a-5p contributed to the continuous suppression of the Th1 response, inducing cytokine expression, IL-17, and IL-2 [59,60]. After *L. (L.) infantum* infection with clinical evolution, dogs were found to produce high Leish-EVs concentrations in blood and high expression of miR21-5p and miR146a-5p [19]. These studies suggest that miR-21-5p and miR-146a-5p can cooperate in the suppression of the protective Th1 response in VL [19,54,55,56,57,58,59,60].

In conclusion, these findings, including this study, originated from in vitro and in vivo analyses and the results are concordant. Here, the THP-1 cells treated with Leish-EVs showed high expression of miR-21-5p and miR-146a-5p and caused the dysregulation of IL-10. Indirectly, these results can suggest that high expression of these miRNAs species is caused by the presence of Leish-EVs. Consequently, this molecular via can contribute to immunosuppression and dysregulation of IL-10, causing enhanced immunopathology in infected hosts.

## Figures and Tables

**Figure 1 microorganisms-12-00270-f001:**
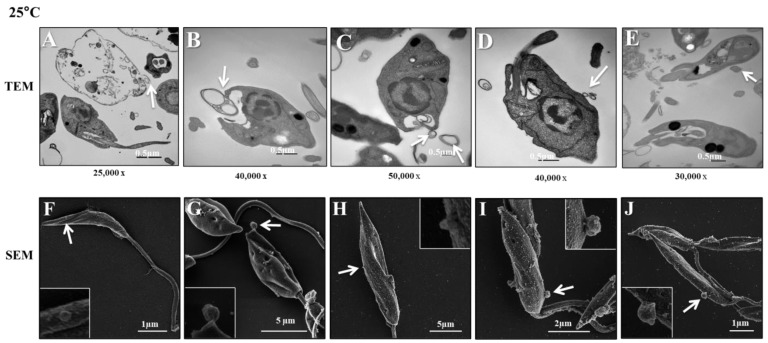
*L. (L.) infantum* releasing Leish-EVs by membrane surface at 25 °C. (**A**–**E**) Images captured through TEM and (**F**–**J**) SEM of promastigotes shedding Leish-EVs by membranes in the beginning of incubation in culture medium at 25 °C (**A**,**F**): after 2 h (**B**,**G**), 4 h (**C**,**H**), 6 h (**D**,**I**), and 24 h (**E**,**J**). The arrows indicate the Leish-EVs. Bars = 2 µm (**F**–**J**).

**Figure 2 microorganisms-12-00270-f002:**
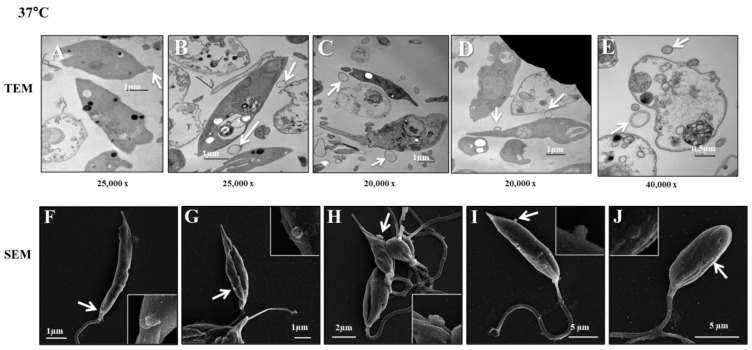
*L. (L.) infantum* releasing Leish-EVs by membrane surface at 37 °C. (**A**–**E**) Images captured through TEM and (**F**–**J**) SEM of promastigotes shedding Leish-EVs by membranes in the beginning of incubation in culture medium at 37 °C (**A**,**F**): after 2 h (**B**,**G**), 4 h (**C**,**H**), 6 h (**D**,**I**), and 24 h (**E**,**J**). The arrows indicate the Leish-EVs. Bars = 2 µm (**F**–**J**).

**Figure 3 microorganisms-12-00270-f003:**
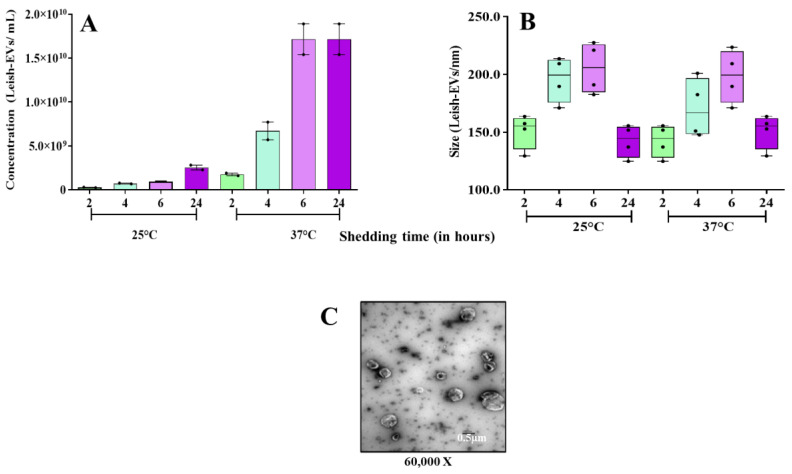
Biological characteristics of Leish-EVs. Distribution of concentration/mL (**A**) and size (nm) (**B**) of Leish-EVs released by 1 × 10^8^ promastigotes incubated in culture medium at 25 °C or 37 °C for 2, 4, 6, and 24 h. The data represent three reads per aliquot and each experiment was performed at least 3 times. TEM image of Leish-EVs released by 1 × 10^8^ promastigotes incubated in culture medium at 25 °C for 2 h with size compatible with microvesicles (**C**).

**Figure 4 microorganisms-12-00270-f004:**
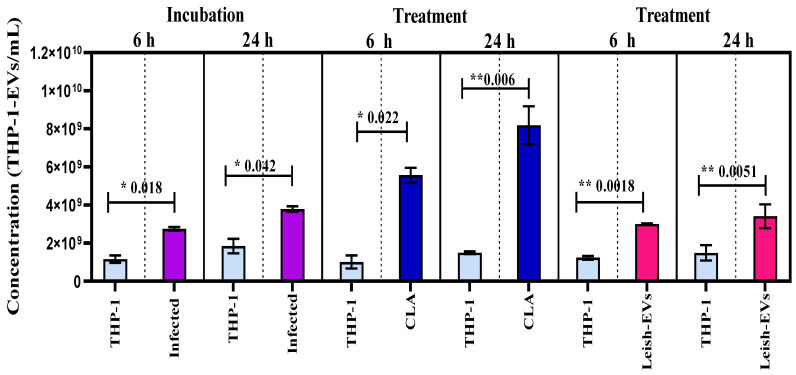
Stimulation of THP-1 cells for releasing EVs. Mean concentration of THP-1-EVs in supernatants analyzed through NTA after incubations during 6 h and 24 h (purple bars) with promastigotes; after CLA treatment (20 µg/well) (dark blue bars); and Leish-EVs (20 µg/well) (dark pink bars). THP-1 cells received only culture medium in controls (light blue bars). The data represent three reads per sample.

**Figure 5 microorganisms-12-00270-f005:**
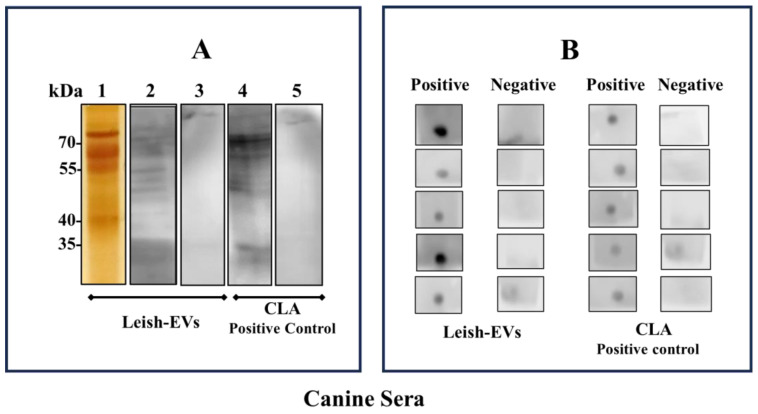
Recognition of Leish-EVs by canine sera. (**A**) Electrophoretic analysis of Leish-EVs on 12% SDS-PAGE stained with silver nitrate (strip 1). Immunoblotting using Leish-EVs as antigen against a positive (strip 2) and negative (strip 3) canine sera for Can-LV. CLA, as positive antigen control, was used against the same sera used for Leish-EVs. Positive (strip 4) and negative (strip 5) sera. (**B**) Dot blot using Leish-EVs as antigen against five positive and five negative sera for Can-LV. CLA was used as positive antigen control.

**Figure 6 microorganisms-12-00270-f006:**
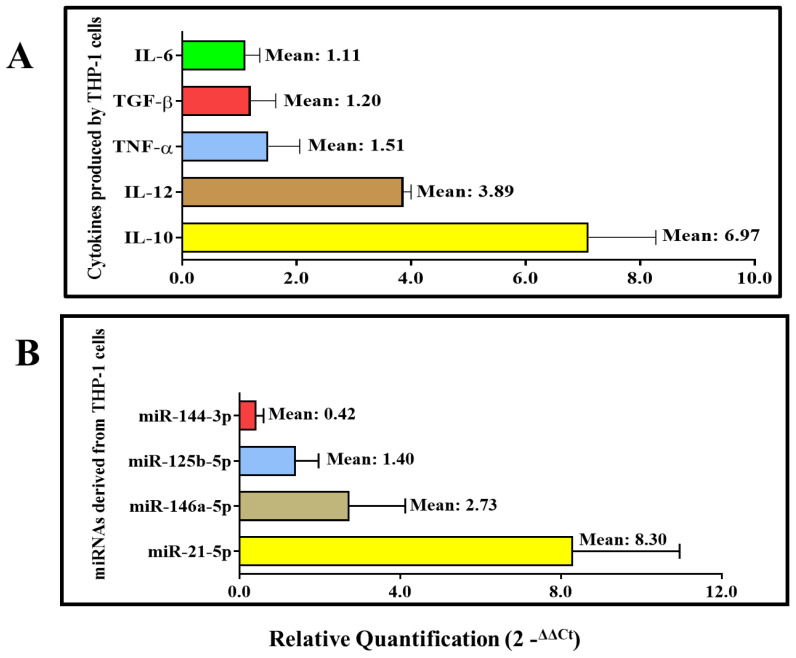
Interaction of THP-1 cells and Leish-EVs. Relative quantifications of the cytokines IL-10, IL-12, TGF-β, TNF-α, IL-6 (**A**), and the miRNAs species miR-21-5p, miR-146a-5p, miR-125b-5p, and miR-144-3p (**B**) in THP-1 cells treated with Leish-EVs. Values are expressed as mean ± SEM of relative quantification after calculation through the comparative CT method (2^−∆∆CT^) as described in the Section 2.

**Table 1 microorganisms-12-00270-t001:** Cytokines assayed in this study.

Gene Symbol	Gene Name	Biological Function	Assay ID	Amplicon Length	Chromosome Location
IFN-γ	Interferon gamma	Protein coding	Hs00989291_m1	73	12
IL10	Interleukin 10	Protein coding	Hs00961622_m1	74	1
IL12	Interleukin 12	Protein coding	Hs01011518_m1	72	5
IL6	Interleukin 6	Protein coding	Hs00985639_m1	66	7
TNF-α	Tumor necrosis factor alpha	Protein coding	Hs01113624_g1	143	6
TGF-β	Transforming growth factor beta 1	Protein coding	Hs00998133_m1	57	19
GAPDH	Glyceraldehyde-3-phosphate dehydrogenase	Protein coding	Hs02758991_g1	93	12

TaqMan real-time PCR assays were chosen to span at least one exon-exon boundary and were purchased as assay IDs from Thermo Fisher Scientific.

**Table 2 microorganisms-12-00270-t002:** miRNAs species assayed in this study.

Assay Name ^1^	Gene Family	Assay ID	Chromosome Localization	Mature miRNA Sequence
1: miR-21-5p	MI0000077	477975_miR	17	UAGCUUAUCAGACUGAUGUUGA
2: miR-146a-5p	MI0000477	478399_miR	5	UGAGAACUGAAUUCCAUGGGUU
3: miR-125b-5p	MIMAT0000423	477885_miR	11	UCC CUG AGA CCC UAA CUU GUGA
4: miR-144-3p	MI0000460	MC11051	17	UACAGUAUAGAUGAUGUACU
5: cel-miR-39-3p ^2^	MIMAT0000010	478293_miR	ND	UCACCGGGUGUAAAUCAGCUUG

ND, not determined; ^1^ purchased from Applied Biosystems; ^2^ external control. Source: http://www.thermofisher.com (accessed on 1 November 2023).

## Data Availability

Data supporting reported results can be requested from the authors.

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
