# Peer review of "Extracellular Vesicles from Leishmania (Leishmania) infantum Contribute in Stimulating Immune Response and Immunosuppression in Hosts with Visceral Leishmaniasis"

_microorganisms, 2024, doi:10.3390/microorganisms12020270_

Round 1

Reviewer 1 Report

Comments and Suggestions for Authors

The paper is very well structured and described.

I appreciated the quality of data presented. 

The data from the results section is clear, and presented in concordance with the aim of the work.

There are  few minor typo:

Eg line 22: 250 C.

Line 49: human e canine must be in english, so replace ,,e„ with „and„.

Line 87, 88, 89, 98,  - add a space.

Figure 5:  the image quality of the blots is low, there seems to be a high degradation of the proteins, therefore a specific band of the right molecular weight (kda) is difficult to be seen. Maybe the authors can capture a better image? In the same manner, maybe the densitometry will be helpful. 

I would recommend adding in conclusion a phrase that captures the impact of the results, both for the diagnosis and the future perspective.

Author Response

-The paper is very well structured and described.

-I appreciated the quality of data presented.

-The data from the results section is clear, and presented in concordance with the aim of the work.

Answer: thank you for your valuable comments.

-There are few minor typo:

-Eg line 22: 250 C.

-Line 49: human e canine must be in English, so replace ,,e„ with „and„.

-Line 87, 88, 89, 98,  - add a space.

Answer: the corrections were performed.

Figure 5:  the image quality of the blots is low, there seems to be a high degradation of the proteins, therefore a specific band of the right molecular weight (kda) is difficult to be seen. Maybe the authors can capture a better image? In the same manner, maybe the densitometry will be helpful.

Answer: We changed the blots. We think that now the images are clearer. Unfortunately, this reaction is not so good because the concentrations of protein is too low in extracellular vesicles.

 -I would recommend adding in conclusion a phrase that captures the impact of the results, both for the diagnosis and the future perspective.

Answer: Based in all comments we modified the objectives and conclusion

Reviewer 2 Report

Comments and Suggestions for Authors

The authors present a study showing the modulation of the immune response induced by EVs from L. infantum. I suggest the authors use the free software Grammarly; the manuscript needs polishing as it has several redactions and grammar mistakes. I also suggest the authors add the recommended reference and further discuss the results.

-The authors should add a quick 2-3 line reading summarizing the study's findings in the last paragraph of the introduction, similar to the abstract.

-Avoid using red and green in the exact figure as it can be challenging to see for vision-impaired people.

-The authors should show in Figure 4 whether there are statistical differences between ‘Infected’ (6 vs 24h) or Treated with CLA or Leish-EVs (6 vs 24h). In addition, the authors should show whether there are statistical differences at the same time point (6 or 24h) in the concentration of THP1 EVs measured after infection, CLA treatment or Leish-EV exposure. This way, the authors can show if there are differences in the optimal time point and type of treatment for THP1 EV secretion.

-Which TNF was evaluated in Figure 6?

-Confirm the mean values of the red and blue bars in Figure 6B. These values are not reflected in the scale. For example, one value is 1.4 but above 2 in the graph.

-The authors reveal in the discussion that IL-10 ‘was able to inhibit the pro-inflammatory cytokines TNF and IL-6’. This is not true, as there are no experiments showing this. You may suggest this based on previous references such as this one (DOI: 10.1016/j.heliyon.2023.e15055). For example, it has been shown that IL-10 blocks the production of IL-12, so why only mention IL-6 and TNF?

-Authors should discuss the miRNA results obtained further. There needs to be more relevance in the discussion about these results. What implications does this have?

Comments on the Quality of English Language

It needs further editing

Author Response

The authors present a study showing the modulation of the immune response induced by EVs from L. infantum. I suggest the authors use the free software Grammarly; the manuscript needs polishing as it has several redactions and grammar mistakes.

Answer: We are grateful for your comments. We checked the manuscript in Grammarly. Thank you.

-I also suggest the authors add the recommended reference and further discuss the results.

-The authors should add a quick 2-3 line reading summarizing the study's findings in the last paragraph of the introduction, similar to the abstract.

Answer: Based in all comments we modified the objectives and conclusion and complement the Discussion section.

-Avoid using red and green in the exact figure as it can be challenging to see for vision-impaired people.

Answer The colors of figures were changed

-The authors should show in Figure 4 whether there are statistical differences between ‘Infected’ (6 vs 24h) or Treated with CLA or Leish-EVs (6 vs 24h). In addition, the authors should show whether there are statistical differences at the same time point (6 or 24h) in the concentration of THP1 EVs measured after infection, CLA treatment or Leish-EV exposure. This way, the authors can show if there are differences in the optimal time point and type of treatment for THP1 EV secretion.

Answer: we calculated all combinations suggested by you. Unfortunately, none of them was statically different.

-Which TNF was evaluated in Figure 6?

Answer:  TNFα. We corrected all TNF-α in manuscript

-Confirm the mean values of the red and blue bars in Figure 6B. These values are not reflected in the scale. For example, one value is 1.4 but above 2 in the graph.

Answer: The figure was changed. We removed the values of the negative control that is 1.0. This information is described in Material and Methods and Results section.

-The authors reveal in the discussion that IL-10 ‘was able to inhibit the pro-inflammatory cytokines TNF and IL-6’. This is not true, as there are no experiments showing this. You may suggest this based on previous references such as this one (DOI: 10.1016/j.heliyon.2023.e15055). For example, it has been shown that IL-10 blocks the production of IL-12, so why only mention IL-6 and TNF?

-Authors should discuss the miRNA results obtained further. There needs to be more relevance in the discussion about these results. What implications does this have?

Answer: Your suggestions were included in the manuscript.

Reviewer 3 Report

Comments and Suggestions for Authors

The manuscript from Carneiro et al., describes how the parasite Leishmania L. infantum released EVs over time and the influence in produce cytokines and microRNA host cells in vitro analysis. L. L. infantum is the most important ethiological agent of visceral leishmaniasis around the world and the studies about the role of EVs show the importance of this multivirulence factor involved in parasite-host interaction.  

It has been described that several Leishmania spp. release EVs containing virulence factors, which may impact the modulation of the cellular immune compartment. 

Respectfully, it is noted that the article lacks novelty regarding the variation in the release of Leishmania vesicles over time and temperature changes. Additionally, it has been demonstrated that some immune responses cells increase the EVs release after stimulation with parasites, antigens or EVs. Furthermore, the group published that increased concentrations of miR-21-5p and miR-146a-5p were significantly higher in the serum of dogs with Can-LV (Da Cruz et al., 2023). 

Despite some limitations, the studies here presented provide important evidence supporting circulating EVs as a potential source of biomarkers and diagnostic antigens in leishmaniasis. In Resume, correct the sentece “cause immunosuppression and production of inflammatory cytokines” to cause a modulation of cytokines production and miRNA. 

 It's worth noting that the studies presented here in vitro data, and caution should be exercised when extrapolating these findings to in vivo systems. Investigations should now be extended to human clinical samples to better understand the communication at the host-Leishmania interface and reveal novel diagnostic markers. 

Major concerns that remain include the following: 

1. Introduction 

Lines 51-54: It is important to note that the current text lacks crucial details. It is apparent that a more comprehensive and detailed explanation of the infection cycle would enhance the content. I recommend further expansion of the infection process, specifically the intricate interactions between the parasite, vector, and host cells. Such additional information would significantly contribute to a more thorough understanding of the complex dynamics at play in this context. 

Lines 69-74: Please, elaborate further on the sentence and include appropriate references. I suggest expanding on the specific details of how Leishmania-derived extracellular vesicles (Leish-EVs) influence the immune response. Some references relevant to the research, such as Zauli et al., 2023; Vasconcelos et al., 2021; Nogueira et al., 2020; and Reis et al., 2020, are still lacking in the current literature. I recommend incorporating these key references to provide a thorough exploration of the topic. This enhancement would contribute to the understanding of the interaction between promastigotes Leish-EVs and the host immune system. 

Authors must significantly improve the manuscript and be more detailed on the role of microRNA present in EVs. Given the group’s specialization in this specific field, including expertise in other diseases, I believe that the importance of microRNA could significantly enhance the overall understanding of the work. 

Line 76: The authors should consider rewriting the sentence: In this case, the EVs were used in vitro studies that showed the recognition by Cani-Leish serum, not stimulation humoral response.   

Materials and Methods  

Lines 159-160: It is not clear if the THP-1 supernatants were submitting to filtration and ultracentrifugation.  

Lines 178: Please, rewrite this part of sentence, changestimulate” for recognized. There are mistakes in the analysis. 

Line 184: Please, add the concentration of CLA and Leish-EV in immunoblotting. 

Line 205: Correct the EVs concentration (20 ug/well) 

Line 222: Add the correct cDNA concentration, not volume 

Lines 246-249: Add the correct samples concentrations, not volume.  

3. Results 

Lines 276: Please, correct tothe ideal in vitro conditions. 

Lines 289-291: The authors should consider standardizing the image sizes by TEM and improving the quality to confirm the difference in EV release between temperatures of 25 and 37 degrees. In my opinion, the figures only indicate a high increase in release over time.  

Figures 1 and 2: Please, standard the arrows size and colors. Indicate the arrow in the correct place. 

Figure 3 Panel A: Indicate the incubation time and temperature. 

Lines 311-318: The authors should consider assessing viability with PI in promastigotes to validate changes in the morphology of the parasites.  

Line 323: It is not clear if the EVs in THP-1 supernatants were separated and concentrated by ultracentrifugation 

Lines 340 and 349: Correct the sentence. It is not the next experiment; it is the same. 

Figure 4: The analysis of the results is long and repetitive. 

Line 358: Immunological experiments  

I wonder why all the 5 serum samples from dogs with CLV were not used in SDS-PAGE and in dot-blot with CLA. Figure 5 should be improved.  

The EVs contents reflect the cell of origin. The authors should consider explaining why the dogs negatively serum was used in SDS-PAGE with EVs and dot-blot if they were test before by ELISA and RT-PCR. 

Figures: Authors should consider improving the layout of the graphics. 

My criticism of this work is that the biological properties of Leish-EVs were assessed in 2 different systems: canine serum and THP-1 cell line from human. there is no data linearity. 

The figure in the attached content is without context and without caption.

4. Discussion 

The logical stream of thought of the discussion does not follow the results. It would be better to reorder. 

Lines 410-411: The discussion in the sentence should be considered exaggerated. I wonder if the correlation does not imply causation; the increased release of EVs at different temperatures may be due to stress. My question is, can macrophage infection with promastigotes be higher with EVs releases at 37 degrees than those at 26? It is interesting how EVs sizes change over time, to both degrees, to 6 and 24 h. What is the author's explanation for this phenomenon? 

Many researchers have demonstrated that EVs isolated from patients can serve as biomarkers during the early clinical stage, acute phase, and chronic phase. The authors should expand their knowledge to include in vivo analysis, especially across different clinical stages.  

Lines 429-430: Please, correct the sentence. The information should be changed to recognized, not stimulate. 

The manuscript should be more accurate. In this case, the study used only 5 serum samples from dogs with Cani-LV, and the results for EVs appear to be like CLA. This data cannot be extrapolated to human LV, and the parasite load should be taken into consideration. 

The EVs structure of L. infantum promastigotes deserves to be further investigated, the presence of glicoconjugates in different temperatures.  

To my opinion, the role of microRNA analysis is still of insufficient quality and does not support the conclusions drawn by the authors  

Lines 439-446: The authors should consider explaining the importance of cytokines and microRNAs in more detail. 

Lines 451-455:  

There are no conclusions in the text. 

A proper in vivo physiological analysis and the investigation of other inflammatory mediators need to be evaluated for more complex discussions and conclusions. The difference in nitric oxide production might reflect differences in pattern recognition receptors activation. This idea would add an interesting topic for the discussion. 

Comments on the Quality of English Language

It is recommended to conduct a thorough review, not only focusing on grammatical aspects but also including a comprehensive examination of the literature and incorporating any necessary updates.

Author Response

The manuscript from Carneiro et al., describes how the parasite Leishmania L. infantum released EVs over time and the influence in produce cytokines and microRNA host cells in vitro analysis. L. L. infantum is the most important etiological agent of visceral leishmaniasis around the world and the studies about the role of EVs show the importance of this multivirulence factor involved in parasite-host interaction. It has been described that several Leishmania spp. release EVs containing virulence factors, which may impact the modulation of the cellular immune compartment.

Respectfully, it is noted that the article lacks novelty regarding the variation in the release of Leishmania vesicles over time and temperature changes. Additionally, it has been demonstrated that some immune responses cells increase the EVs release after stimulation with parasites, antigens or EVs.  Furthermore, the group published that increased concentrations of miR-21-5p and miR-146a-5p were significantly higher in the serum of dogs with Can-LV (Da Cruz et al., 2023).

Despite some limitations, the studies here presented provide important evidence supporting circulating EVs as a potential source of biomarkers and diagnostic antigens in leishmaniasis.

Answer: thank you for your valuable comments.

In Resume correct the sentence “cause immunosuppression and production of inflammatory cytokines” to cause a modulation of cytokines production and miRNA.

Answer: the corrections were performed.

 It's worth noting that the studies presented here in vitro data, and caution should be exercised when extrapolating these findings to in vivo systems. Investigations should now be extended to human clinical samples to better understand the communication at the host-Leishmania interface and reveal novel diagnostic markers.

Answer: Thanks for your observations. However, our group studied these data that produced this manuscript based in our previous study (Cruz et al. 2023), in which serum samples before and after infection with L. (L.) infantum were judiciously analyzed. In this way, based in difficult to understand some phenomena in vivo we decided to produce this study.

Major concerns that remain include the following:

Introduction

Lines 51-54: It is important to note that the current text lacks crucial details. It is apparent that a more comprehensive and detailed explanation of the infection cycle would enhance the content. I recommend further expansion of the infection process, specifically the intricate interactions between the parasite, vector, and host cells. Such additional information would significantly contribute to a more thorough understanding of the complex dynamics at play in this context.

Lines 69-74: Please, elaborate further on the sentence and include appropriate references. I suggest expanding on the specific details of how Leishmania-derived extracellular vesicles (Leish-EVs) influence the immune response. Some references relevant to the research, such as Zauli et al., 2023; Vasconcelos et al., 2021; Nogueira et al., 2020; and Reis et al., 2020, are still lacking in the current literature. I recommend incorporating these key references to provide a thorough exploration of the topic. This enhancement would contribute to the understanding of the interaction between promastigotes Leish-EVs and the host immune system.

Lines 69-74: Please, elaborate further on the sentence and include appropriate references. I suggest expanding on the specific details of how Leishmania-derived extracellular vesicles (Leish-EVs) influence the immune response. Some references relevant to the research, such as; and Reis et al., 2020, are still lacking in the current literature. I recommend incorporating these key references to provide a thorough exploration of the topic. This enhancement would contribute to the understanding of the interaction between promastigotes Leish-EVs and the host immune system.

Authors must significantly improve the manuscript and be more detailed on the role of microRNA present in EVs. Given the group’s specialization in this specific field, including expertise in other diseases, I believe that the importance of microRNA could significantly enhance the overall understanding of the work.

Line 76: The authors should consider rewriting the sentence: In this case, the EVs were used in vitro studies that showed the recognition by Cani-Leish serum, not stimulation humoral response.  

Answer: We rewrote all Introduction item based in your comments

Materials and Methods 

Lines 159-160: It is not clear if the THP-1 supernatants were submitting to filtration and ultracentrifugation.

Answer: The total supernatants were directly analyzed without any treatment. The information was included in Material and Methods section.

Lines 178: Please, rewrite this part of sentence, change “stimulate” for recognized. There are mistakes in the analysis.

Answer: The modification was performed.

Line 184: Please, add the concentration of CLA and Leish-EV in immunoblotting.

Answer: The concentration of both (CLA and Leish-EV) were included

Line 205: Correct the EVs concentration (20 ug/well)

Answer: The concentration was corrected

Line 222: Add the correct cDNA concentration, not volume

Lines 246-249: Add the correct samples concentrations, not volume. 

Answer: According TaqMan Advanced miRNA cDNA synthesis kit (Applied Biosystems); Customized assay by Applied Biosystems and TaqMan Fast Advanced Master Mix, the correct values must be in volume and not in concentration

  1. Results

Lines 276: Please, correct to “the ideal in vitro conditions”.

Answer: The correction was performed

-Lines 289-291: The authors should consider standardizing the image sizes by TEM and improving the quality to confirm the difference in EV release between temperatures of 25 and 37 degrees. In my opinion, the figures only indicate a high increase in release over time.

-Figures 1 and 2: Please, standard the arrows size and colors. Indicate the arrow in the correct place.

Answer: The corrections in figures was performed

Figure 3 Panel A: Indicate the incubation time and temperature.

Answer: The information was included

Lines 311-318: The authors should consider assessing viability with PI in promastigotes to validate changes in the morphology of the parasites. 

Answer: We performed at least 3 times the same experiment and the results always were similar. This information was included in each experiment.

Line 323: It is not clear if the EVs in THP-1 supernatants were separated and concentrated by ultracentrifugation. 

Answer: We already informed the methodology in Material and Methods section. The supernatants were analyzed without concentration by ultracentrifugation. The cells produced a good quantity of EVs and it was not necessary any concentration.

Lines 340 and 349: Correct the sentence. It is not the next experiment; it is the same.

Answer: We removed the “next”

Figure 4: The analysis of the results is long and repetitive.

Answer:  the text has been summarized.

Line 358: Immunological experiments 

I wonder why all the 5 serum samples from dogs with CLV were not used in SDS-PAGE and in dot-blot with CLA. Figure 5 should be improved. 

Answer: Our objective in this part of the study was to investigate whether infected hosts produce EVs against L. (L.) infantum. Any mammal produces antibodies against EVs from parasites (and other microorganisms).

As described in Material and Methods section and Figures, CLA and live promastigotes were used as positive controls of the reactions. As we have many Dot-Blot results in dog sera we include 5 control sera in Figure 5B.

-The EVs contents reflect the cell of origin. The authors should consider explaining why the dogs negatively serum was used in SDS-PAGE with EVs and dot-blot if they were test before by ELISA and RT-PCR.

Answer: This information is in Material and Methods section, Item 2.6. Immunological investigations, line 6th ...The canine sera have been previously tested for Can-VL by ELISA, Fast test, and Real-Time PCR .

-Figures: Authors should consider improving the layout of the graphics.

Answer: All figures were improved  

-My criticism of this work is that the biological properties of Leish-EVs were assessed in 2 different systems: canine serum and THP-1 cell line from human. there is no data linearity.

The figure in the attached content is without context and without caption.

Answer: As we informed  above any mammal produces antibodies against EVs from parasites.  This information is important principally for use of EVs as biomarkers

As described in Material and Methods section and Figures, CLA and live promastigotes were used as positive controls of the reactions. As we have many Dot-Blot results in dog sera we include 5 control sera in Figure 5B.

  1. Discussion

-The logical stream of thought of the discussion does not follow the results. It would be better to reorder.

Lines 410-411: The discussion in the sentence should be considered exaggerated. I wonder if the correlation does not imply causation; the increased release of EVs at different temperatures may be due to stress. My question is, can macrophage infection with promastigotes be higher with EVs releases at 37 degrees than those at 26? It is interesting how EVs sizes change over time, to both degrees, to 6 and 24 h. What is the author's explanation for this phenomenon?

Answer: Sorry, but the information is not exaggerated. The same data already were described by others, analyzing other Leishmania species. Please see in reference section. Regarding EV size, the data showed that in all periods the parasites released only microvesicles. No difference was shown.

-Many researchers have demonstrated that EVs isolated from patients can serve as biomarkers during the early clinical stage, acute phase, and chronic phase. The authors should expand their knowledge to include in vivo analysis, especially across different clinical stages. 

-Lines 429-430: Please, correct the sentence. The information should be changed to recognized, not stimulate.

-The manuscript should be more accurate. In this case, the study used only 5 serum samples from dogs with Cani-LV, and the results for EVs appear to be like CLA. This data cannot be extrapolated to human LV, and the parasite load should be taken into consideration.

The EVs structure of L. infantum promastigotes deserves to be further investigated, the presence of glicoconjugates in different temperatures. 

Answers: Lines 429-430. We corrected de word. In this paragraph we include the information that we already evaluated EVs produced by hosts with V (Cruz et al., 2023).

The study of glicoconjugates must be performed by other researcher groups. This area is out of my interest. On the other hand, the immune system of hosts having VL produce antibodies against different parasite molecules

-To my opinion, the role of microRNA analysis is still of insufficient quality and does not support the conclusions drawn by the authors.  

-Lines 439-446: The authors should consider explaining the importance of cytokines and microRNAs in more detail.

-Lines 451-455: 

-There are no conclusions in the text.

-A proper in vivo physiological analysis and the investigation of other inflammatory mediators need to be evaluated for more complex discussions and conclusions. The difference in nitric oxide production might reflect differences in pattern recognition receptors activation. This idea would add an interesting topic for the discussion.

Answers: The modifications and inclusions were performed

Round 2

Reviewer 3 Report

Comments and Suggestions for Authors

Major concerns that remain include the following: 

Line 74-75: It is repetitive. 

Line 78: Correct the distinct cytokines (IL-10 is an immunoregulatory cytokine) 

Lines 84-86: The sentence should be revised. 

Lines 108-147: The sentence should be revised, it is repeated. 

Lines 149: The claim “contribute to immunosuppression and production of inflammatory cytokines Surprisingly, only a high production of IL-10 seems to be significantly different from value for negative cells.  

Results 

Biological characteristics of Leish-EVs released by L. (L.) infantum: In my opinion, figures 1 and 2 only indicate a high increase in release over time  

Figures 1 and 2: Please, indicate the bars in the correct place and all images. 

Figure 4. Correct the title. It is notRecognition”. 

-Immunological experiments 

I wonder why the Immunological experiments were using canine sera intead of human sera? 

Line 523: It seems that the authors did not understand the difference between recognition (reactivity) and stimulation. The sentence should be revised. 

Lines 631: The main cell producer of IFN-y is the NK cells, macrophages do not produce significant amounts of IFN-y. The authors can confirm that in other papers. 

-Discussion 

Lane 618: In this study, the analysis was performed using canine sera, so the immunoblots and dot-blots can be used as biomarkers in diagnosis for Can-VL. 

Proper physiological analysis and the investigation of other inflammatory mediators need to be evaluated for more complex discussions and conclusions. 

Comments on the Quality of English Language

It is important that the manuscript undergoes English review before being considered for publication.  

Some parts of text are confusing.

Author Response

Major concerns that remain include the following:

-Line 74-75: It is repetitive.

-Line 78: Correct the distinct cytokines (IL-10 is an immunoregulatory cytokine)

-Lines 84-86: The sentence should be revised.

-Lines 108-147: The sentence should be revised, it is repeated.

-Lines 149: The claim “contribute to immunosuppression and production of inflammatory cytokines” Surprisingly, only a high production of IL-10 seems to be significantly different from value for negative cells. 

Answer: All considerations were revised and reformulated. The text was, also, revised. We, also verified concordance errors. Thank you for your consideration.

Results

-Biological characteristics of Leish-EVs released by L. (L.) infantum: In my opinion, figures 1 and 2 only indicate a high increase in release over time.  

Answer: We included in Results section the information according Figures 1, 2 and 3.

-Figures 1 and 2: Please, indicate the bars in the correct place and all images.

Answer: We changed the figures as you suggested.

Figure 4. Correct the title. It is not “Recognition”.

Answer: We change the title of figure and the text.

-Immunological experiments

-I wonder why the Immunological experiments were using canine sera intead of human sera?

-Line 523: It seems that the authors did not understand the difference between recognition (reactivity) and stimulation. The sentence should be revised.

-Lines 631: The main cell producer of IFN-y is the NK cells, macrophages do not produce significant amounts of IFN-y. The authors can confirm that in other papers.

-Discussion

-Lane 618: In this study, the analysis was performed using canine sera, so the immunoblots and dot-blots can be used as biomarkers in diagnosis for Can-VL.

-Proper physiological analysis and the investigation of other inflammatory mediators need to be evaluated for more complex discussions and conclusions.

Answer: Thank you for your valuable comments. These questions were included in Results or Discussion sections. We included more two articles in References according comments of line 631.